# Knowledge and preventive practices towards COVID-19 among pregnant women seeking antenatal services in Northern Ghana

Maxwell Tii Kumbeni[1]*, Paschal Awingura Apanga[2], Eugene Osei Yeboah[3], Isaac Bador Kamal Lettor[4]

**1** Ghana Health Service, Nabdam District Health Directorate, Nangodi, Ghana, **2** School of Community Health Sciences, University of Nevada, Reno, Reno, Nevada, United States of America, **3** Ghana Health Service, Bolgatanga East District Health Directorate, Zuarungu, Ghana, **4** Bawku Technical Institute, Bawku, Upper East Region, Ghana

* tiimax2@gmail.com

## Abstract

**Data Availability Statement:** All relevant data are available within the paper and its Supporting information files.

### Background

COVID-19 is a novel respiratory disease associated with severe morbidity and high mortality in the elderly population and people with comorbidities. Studies have suggested that pregnant women are more susceptible to COVID-19 compared to non-pregnant women. However, it's unclear whether pregnant women in Ghana are knowledgeable about COVID-19 and practice preventive measures against it. This study sought to assess the knowledge and preventive practices towards COVID-19 among pregnant women seeking antenatal services in Northern Ghana.

### Methods

A cross-sectional study was conducted using a structured questionnaire in the Nabdam district in Ghana. A total of 527 pregnant women were randomly sampled from health facilities offering antenatal care services in the district. Multivariable logistic regression analysis was used to assess the association between the predictors and outcome variables.

### Results

The prevalence of adequate knowledge and good COVID-19 preventive practices were 85.6%, (95% CI: 82.57, 88.59) and 46.6%, (95% CI: 42.41, 50.95) respectively. Having at least a primary education, residing in an urban area, and receiving COVID-19 education at a health facility were positively associated with adequate knowledge on COVID-19. Factors positively associated with good COVID-19 preventive practices were older age, having at least a primary education, pregnant women with a chronic disease, and living in an urban area. Multiparity was negatively associated with good COVID-19 preventive practices.

**Funding:** The authors received no specific funding for this work.

**Competing interests:** The authors have declared that no competing interests exist.

## Conclusion

Although majority of women had adequate knowledge of COVID-19, less than half of them were engaged in good COVID-19 preventive practices. Education of pregnant women on COVID-19 preventive practices should be intensified at health facilities while improving upon the water, sanitation and hygiene need particularly in rural communities.

## Introduction

Coronavirus disease 2019 (COVID-19) is an emerging respiratory disease caused by severe acute respiratory syndrome coronavirus-2 (SARS-COV2), which is a single-strand, positive-sense ribonucleic acid (RNA) virus [1]. Confirmed cases of COVID-19 usually present with clinical signs and symptoms of fever, dry cough, tiredness, and shortness of breath with an incubation period of 2–14 days after exposure to the virus [2–5]. The virus may cause morbidity in the range of mild respiratory illness to severe complications characterized by acute respiratory distress syndrome, septic shock, and other metabolic and hemostasis disorders, and eventually death [4, 5]. Most of the fatal forms of COVID-19 including acute respiratory syndrome occurred in older adults and people with underlying medical comorbidities [6–8]. A systematic review by Yang et al., found that individuals with hypertensive, cardiovascular, and respiratory system diseases were the most vulnerable groups associated with mortality due to COVID-19 [9].

As the outbreak of COVID-19 continues to unfold, major concerns are being raised about its effects on pregnancy and the potential risk of vertical transmission. Recent evidence suggests that the risk of maternal mortality appears to be high in COVID-19 pregnant women with severe disease [10]. There is limited evidence on intra-uterine transmission of COVID-19 from mother to child [11, 12]. Whilst some newborns have tested negative for COVID-19 after birth, some have tested positive after few days of life [13]. It is however unclear at what stage (pre, peri, or postnatal) the transmission might have occurred among newborns who tested positive [14]. In early studies from China, it was observed that some newborns were preterm and low birth weight when born to COVID-19 positive mothers, but the evidence linking these outcomes to the COVID-19 is unclear [15]. Although the impact of COVID-19 on pregnant women is not yet known, there is the need to consider pregnant women as a high-risk population in COVID-19 prevention and control strategies [16, 17].

Although vaccines for COVID-19 are now available, it is not clear if vaccines can prevent transmission of the virus [18]. Therefore, practicing COVID-19 preventive measures is critical in the control of the COVID-19 pandemic [19]. Accordingly, various interventions have been implemented globally such as partial lockdowns, contact tracing, and self-isolation or quarantine, and promotion of public health measures including hand hygiene, respiratory protocols, and social distancing to curb the spread of the virus [20].

Ghana reported its first case of COVID-19 on 12[th] March 2020 in its national capital, Accra [21]. Cases of COVID-19 have since spread to all regions of the country, and as of 8[th] April 2021, Ghana has recorded a total of 91,109 cases and 752 deaths [22]. Ghana has adopted several measures to fight the virus namely; testing, tracing, and treating, the partial lockdown of some major cities, and practicing COVID-19 safety measures [23]. It has also resorted to the use of geospatial technology in its effort to enhance contact tracing and improving decision making [24]. Successful control of COVID-19 infection will require a change of individual behavior, and this is influenced by people's understanding of the characteristics of the disease and its preventive measures [25]. Studies on knowledge and preventive practices in Ghana are

focused on health workers and the general public [26–28], but not pregnant women. Therefore, the study assessed the knowledge and preventive practices of pregnant women towards COVID-19 in the Nabdam district of Ghana.

## Materials and methods

### Study design, population, and setting

A population-based cross-sectional survey was conducted in health facilities using a quantitative approach. The study involved pregnant women who were 18 years and above, and accessed antenatal care services in the Nabdam District of Upper East Region, Ghana. The survey was conducted in October 2020.

### Sample size and sampling procedure

The sample size for the study was estimated using EpiInfo Version 7.1 (STAT CALC). The prevalence of knowledge on COVID-19 was not known, therefore a 50% prevalence rate was used with a 95% confidence interval and 5% margin of error. The estimated minimum sample size was 407 including a 10% non-response rate. However, 527 participants completed our surveys. A pretest was conducted to ascertain the validity of the questionnaire.

A total of 16 health facilities that conduct antenatal care services in the district were purposively selected for the study. The selection of study participants from the health facilities was done using a simple random sampling method. Face-to-face interviews were conducted with a structured questionnaire.

### Predictor variables

The predictor variables were age, parity, marital status, educational level, gestational age, has a chronic disease, the number of antenatal visits, residential area, and health education on COVID-19 at a health facility. These variables were categorized as age (18–22 years, 23-27years, $\geq$ 28 years); parity (0, 1, 2+); marital status (never married, married); educational level (no formal education, primary education, secondary or higher education); gestational age (first trimester, second trimester, third trimester); residential area (rural, urban); the number of antenatal visits (1–3, 4+); received COVID-19 education at a health facility (yes, no); and has a chronic disease (yes, no). Having a chronic disease was defined as having any one of the following diseases; hypertension, diabetes, sickle cell disease, asthma, cancer, chronic obstructive pulmonary disease.

### Outcome variables

We assessed two outcome variables; knowledge of COVID-19 and COVID-19 preventive practices. Knowledge was assessed on a 10-item questionnaire adapted from Ah-Hanawi et al., [28], and the level of knowledge was adapted from Bloom's cut-off point [29]. The questions were about clinical presentations, transmission, prevention, and control of COVID-19. Each correct response weighed 1 point and 0 for an incorrect response. A score of 6 points and above was considered adequate knowledge while 5 points and below was considered inadequate knowledge. Questions used to construct our outcome variable on knowledge of COVID-19 are shown in S1 Appendix.

COVID-19 preventive practices were assessed on a 5-item questionnaire derived from W.H.O. recommendations on preventive measures against COVID-19 [30] and cut-off points adapted from Bloom's [29]. Each correct response weighed 1 point and 0 for an incorrect response. A score of 3 points and above was considered good COVID-19 preventive practices

while 2 points and below was considered poor COVID-19 preventive practices. S2 Appendix also shows questions that were used to define COVID-19 preventive practices.

### Data processing and analysis

Data analysis was done using SAS version 9.3 (SAS Institute, Cary, NC). Descriptive statistics were used to present the characteristics of study participants. Multivariable logistic regression analysis was used to assess the association between the predictors and outcome variables, while simultaneously controlling for predictor variables. A P-value <0.05 was considered statistically significant.

### Ethical consideration

Approval was obtained from the Committee on Human Research, Publication, and Ethics at the School of Medical Sciences /Komfo Anokye Teaching Hospital (CHRPE/AP/369/20). Permission was also sought from the district management as well as heads of the various health facilities. Written informed consent was obtained from all the study participants.

## Results

### Characteristics of the study sample

The ages of participants were fairly distributed between pregnant women aged 18–22 years old (37.0%) and above 28 years old (36.1%). Pregnant women with more than one child formed the majority of the participants (44.0%). The proportion of pregnant women with a primary education was 46.9%, secondary or higher education (30.5%), and no formal education (22.6%). The majority of the pregnant women were in their second trimester (41.0%), and most (91.4%) had no chronic diseases. Women who made one to three antenatal care visits were 45.8% while four visits and above were 54.2%. More than half (65.6%) of the women had received COVID-19 education from a health facility (Table 1).

### Prevalence of knowledge and COVID-19 preventive practices

More than two-thirds of the participants had adequate knowledge of COVID-19, 85.6% (95% CI: 82.57, 88.59). However, less than half of them were found to be engaged in good COVID-19 preventive practices, 46.6% (95% CI: 42.41, 50.95) [Fig 1].

### Factors associated with knowledge on COVID-19 and COVID-19 preventive practices

Pregnant women with a primary education [Adjusted prevalence odds ratios (aOR): 3.40, 95% CI: 1.79, 6.46], and secondary or higher education (aOR: 10.61, 95% CI: 3.59, 31.33), had 3.40 times and 10.61 times respectively, the odds of having adequate knowledge on COVID-19 compared to those with no formal education. Women residing in urban areas had 119% higher odds of having adequate knowledge on COVID-19 compared to those living in rural areas (aOR: 2.19, 95% CI: 1.11, 4.35). Women who received COVID-19 education at a health facility had 112% higher odds of having adequate knowledge on COVID-19 compared to women who did not receive COVID-19 education at a health facility (aOR: 2.12, 95% CI: 1.18, 3.82). All other variables were not associated with knowledge on COVID-19 (Table 2).

The odds of engaging in good COVID-19 preventive practices among women aged 23–27 years (aOR: 1.85, 95% CI: 1.04, 3.31), and 28 years and above (aOR: 2.12, 95% CI: 1.06, 4.23) were 1.85 times and 2.12 times respectively, the odds of good COVID-19 preventive practices compared to women aged 18–22 years old. Multiparous (i.e. more than 1 child) women had

**Table 1. Characteristics of the study sample (n = 527).**

| Variable | N (%) |
|---|---|
| **Age (years)** | |
| 18–22 | 195 (37.0) |
| 23–27 | 142 (26.9) |
| > 28 | 190 (36.1) |
| **Parity** | |
| 0 | 168 (31.9) |
| 1 | 127 (24.1) |
| 2+ | 232 (44.0) |
| **Marital status** | |
| Never married | 81 (15.4) |
| Married | 446 (84.6) |
| **Educational level** | |
| No formal education | 119 (22.6) |
| Primary education | 247 (46.9) |
| Secondary or higher education | 161 (30.5) |
| **Gestational age** | |
| First trimester | 128 (24.3) |
| Second trimester | 216 (41.0) |
| Third trimester | 183 (34.7) |
| **Has a chronic disease** | |
| No | 481 (91.4) |
| Yes | 45 (8.6) |
| **Residential area** | |
| Rural | 306 (58.1) |
| Urban | 221 (41.9) |
| **Number of antenatal visits** | |
| 1–3 | 241 (45.8) |
| 4+ | 285 (54.2) |
| **Received COVID-19 education at a health facility** | |
| No | 179 (34.4) |
| Yes | 341 (65.6) |

54% lower odds of engaging in good COVID-19 preventive practices compared to nulliparous women (aOR: 0.46, 95% CI: 0.22, 0.95). Women with primary education (aOR: 2.11, 95% CI: 1.22, 3.64), and secondary or higher education (aOR: 4.11, 95% CI: 2.18, 7.74), had 2.11 times and 4.11 times respectively, the odds of engaging in good COVID-19 preventive practices compared to women with no formal education. The odds of engaging in good COVID-19 preventive practices were 111% higher in women who had a chronic condition compared to women without a chronic condition (aOR: 2.11, 95% CI: 1.09, 4.11), and women living in urban areas had 89% higher odds of engaging in good COVID-19 preventive practices compared to those in rural areas (aOR: 1.89, 95% CI: 1.26, 2.83). All other variables were not associated with COVID-19 preventive practices (Table 2).

## Discussion

We investigated the knowledge and preventive practices towards COVID-19 among pregnant women seeking antenatal services in Ghana. Our study found that more than 8 in every 10

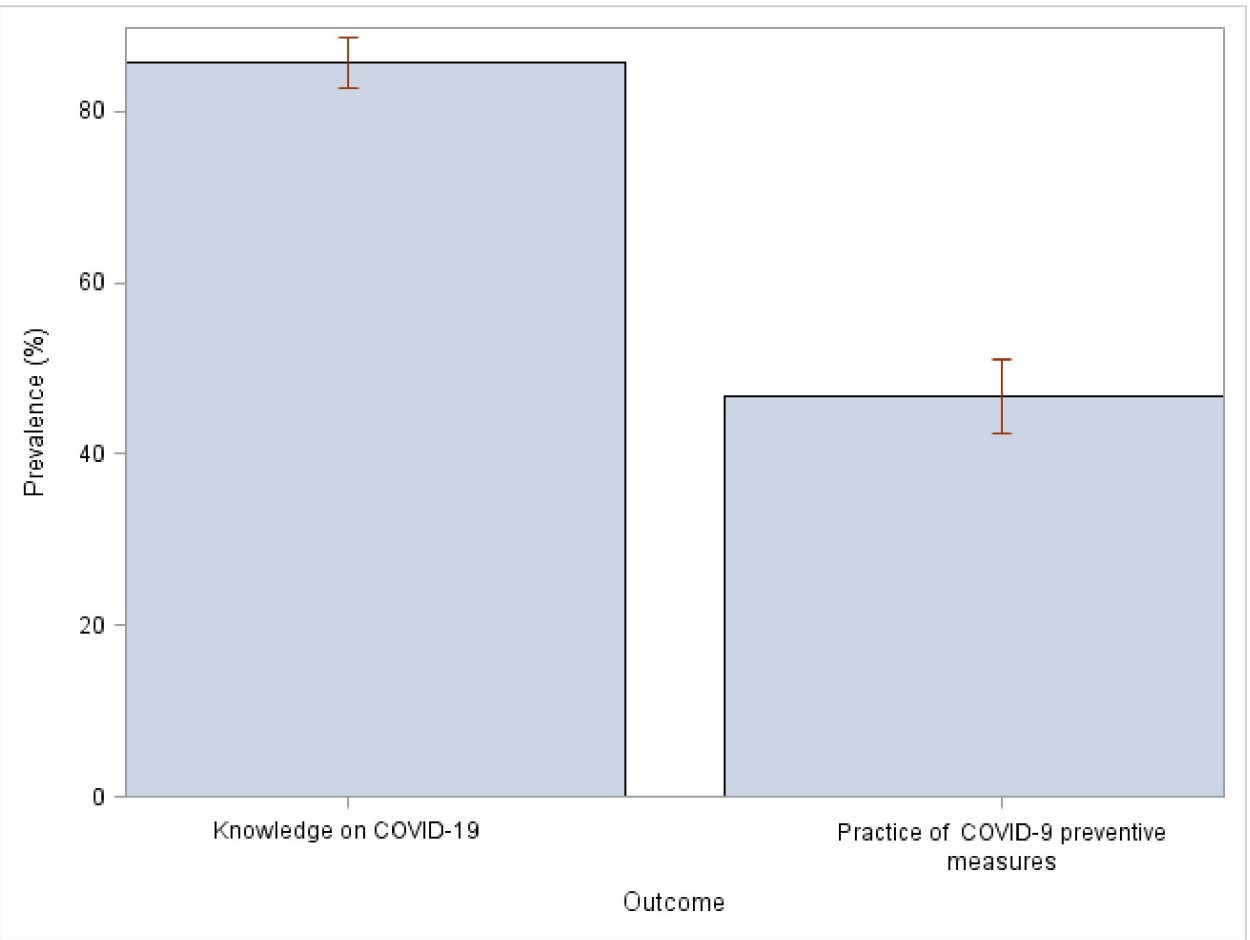

**Fig 1. Prevalence of knowledge and COVID-19 preventive practices.**

pregnant women had adequate knowledge of COVID-19, however, less than half of the participants were engaged in good COVID-19 preventive practices. Having at least primary education, residing in an urban area, and women who received COVID-19 education at a health facility were positively associated with adequate knowledge on COVID-19. Good COVID-19 preventive practices were more prevalent among older pregnant women, pregnant women who had at least primary education, pregnant women who had a chronic disease, and those living in an urban area. Multiparity was negatively associated with good COVID-19 preventive practices.

Our study found that majority of the women had adequate knowledge of COVID-19, but less than half of the women were engaged in good COVID-19 preventive practices. This finding is corroborated in a similar study by Nwafor et al., in Nigeria [31]. The high prevalence of adequate knowledge may be attributed to intense information sharing through media and other platforms. For example, since the emergence of the disease, there has been constant sharing of information on COVID-19 by the Government of Ghana, civil society organizations, and individuals via social media, television, radio, and mobile van announcements [32]. Health education on COVID-19 has also been ongoing at the various health facilities. The low prevalence of good preventive practices could be due to several reasons. Inadequate face masks, challenges with water supply systems, and unavailability of soap for handwashing as well as the

**Table 2. Factors associated with knowledge on COVID-19 and practice of COVID-19 preventive measures.**

| Variable | Knowledge of COVID-19 | COVID-19 preventive practices |
|---|---|---|
| | Adjusted OR (95% CI) | Adjusted OR (95% CI) |
| **Age (years)** | | |
| 18–22 | 1 | 1 |
| 23–27 | 1.31 (0.50, 3.44) | 1.85 (1.04, 3.31) * |
| ≥ 28 | 1.21 (0.44, 3.28) | 2.12 (1.06, 4.23) * |
| **Parity** | | |
| 0 | 1 | 1 |
| 1 | 0.84 (0.31, 2.28) | 0.77 (0.44, 1.34) |
| 2+ | 0.42 (0.15, 1.17) | 0.46 (0.22, 0.95) * |
| **Marital status** | | |
| Never married | 1 | 1 |
| Married | 0.97 (0.33, 2.91) | 1.53 (0.88, 2.64) |
| **Educational level** | | |
| No formal education | 1 | 1 |
| Primary education | 3.40 (1.79, 6.46) * | 2.11 (1.22, 3.64) * |
| Secondary or higher education | 10.61 (3.59, 31.33) * | 4.11 (2.18, 7.74) * |
| **Gestational age** | | |
| First trimester | 1 | 1 |
| Second trimester | 1.10 (0.55, 2.18) | 0.72 (0.43, 1.23) |
| Third trimester | 0.81 (0.32, 2.08) | 0.87 (0.45, 1.69) |
| **Has a chronic disease** | | |
| No | 1 | 1 |
| Yes | 2.72 (0.90, 8.28) | 2.11 (1.09, 4.11) * |
| **Residential area** | | |
| Rural | 1 | 1 |
| Urban | 2.19 (1.11, 4.35) * | 1.89 (1.26, 2.83) * |
| **Number of antenatal visits** | | |
| 1–3 | 1 | 1 |
| 4+ | 0.98 (0.46, 2.10) | 1.25 (0.75, 2.06) |
| **Received COVID-19 education at a health facility** | | |
| No | 1 | 1 |
| Yes | 2.12 (1.18, 3.82) * | 1.22 (0.80, 1.87) |

*P-value is less than 0.05.

high cost of hand sanitizers might be the possible explanation for the low level of adherence to COVID-19 preventive practices [33].

Women who had at least primary education and women living in urban areas were positively associated with adequate knowledge on COVID-19. Our finding is supported by Nwafor et al. [31]. Having at least primary education is often associated with easier access to health information compared to individuals without a formal education [34], and this might explain why women with primary or higher education had adequate knowledge on COVID-19. Most educated pregnant women live in the urban areas [35], more so, the urban areas have good infrastructure such as internet connectivity and other media facilities compared to the rural areas, and this may account for the high prevalence of adequate knowledge on COVID-19 among women living in urban areas [36]. We also found that women who received COVID-19 education at a health facility had adequate knowledge on COVID-19 compared to those who

did not receive COVID-19 education at a health facility. This confirms the key role that health workers play in disseminating information regarding COVID-19 at health facilities.

Although our study found a low prevalence of good COVID-19 preventive practices, older age women were positively associated with good COVID-19 preventive practices. Women who were 28 years old and above were more likely to engage in good COVID-19 preventive practices compared to women aged 18–22 years old. Studies have shown that older age is a risk factor for severe complications and fatality related to COVID-19 [7, 8]. This might be the reason why older women in our study were more engaged in good COVID-19 preventive practices to avoid getting infected with the disease. Pregnant women with at least a primary education were also associated with good COVID-19 preventive practices. Women with at least a primary education may be more exposed to health information especially regarding COVID-19 and are therefore likely to take positive measures to protect themselves against the disease [37].

We also found that women who had a chronic disease were more likely to engage in good COVID-19 preventive practices compared to those who have no chronic disease. As has been found in most COVID-19 studies, severe complications and fatalities occur in people with chronic diseases such as hypertension, diabetes, and respiratory chronic diseases [7–9]. It is therefore not surprising that women with such chronic diseases took more precautions in protecting themselves against COVID-19 compared to women without chronic disease. Good COVID-19 preventive practices were also associated with living in an urban area. Women living in urban areas had higher odds of engaging in good COVID-19 preventive practices compared to those in rural areas. In Ghana, the prevalence of COVID-19 has been high in the urban areas compared to the rural areas [38], and this might have been the reason why pregnant women in urban areas are more engaging in good COVID-19 preventive practices. Our study also found that multiparous women had lower odds of engaging in good COVID-19 preventive practices. This finding might be because multiparous women are associated with lower education and mostly reside in rural areas [39].

Our study had some strengths and limitations. Our findings are relevant to inform policymakers in channeling resources towards the fight against COVID-19. The data was self-reported and might have suffered recall bias. Also, the cross-sectional nature of our data does not allow for our findings to infer causality. Our findings may not be generalized to the entire country however, our findings are useful and are the first to assess the level of knowledge and preventive practices of pregnant women towards COVID-19 in Ghana.

## Conclusion

Although knowledge on COVID-19 among pregnant women was high, this did not reflect into pregnant women engaging in good COVID-19 preventive practices. There is a need to institute measures to improve COVID-19 preventive practices among pregnant women in Ghana. One of the ways of achieving this is by extending the media campaign to rural areas, where access to electronic media is limited. Also, efforts should be made to improve water, sanitation, and hygiene systems in communities as well as the free supply of facemask to the underprivileged.

## Supporting information

**S1 Appendix. Questions on knowledge of COVID-19.**
(DOCX)

**S2 Appendix. Questions on COVID-19 preventive practices.**
(DOCX)

**S1 Dataset.**
(XLSX)

## Acknowledgments

We acknowledge Mr. Baba Awuni, Ms. Augustina Yenlokre, and Mr. Richard Sodana for their contribution. We also acknowledge all our participants.

## Author Contributions

**Conceptualization:** Maxwell Tii Kumbeni, Paschal Awingura Apanga.

**Data curation:** Maxwell Tii Kumbeni, Paschal Awingura Apanga, Eugene Osei Yeboah, Isaac Bador Kamal Lettor.

**Formal analysis:** Paschal Awingura Apanga.

**Methodology:** Maxwell Tii Kumbeni, Eugene Osei Yeboah, Isaac Bador Kamal Lettor.

**Project administration:** Maxwell Tii Kumbeni.

**Visualization:** Maxwell Tii Kumbeni, Paschal Awingura Apanga, Eugene Osei Yeboah, Isaac Bador Kamal Lettor.

**Writing – original draft:** Maxwell Tii Kumbeni, Paschal Awingura Apanga, Eugene Osei Yeboah, Isaac Bador Kamal Lettor.

**Writing – review & editing:** Maxwell Tii Kumbeni, Paschal Awingura Apanga, Eugene Osei Yeboah, Isaac Bador Kamal Lettor.

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
