## [Decision Letter · Decision Letter 0]

18 Feb 2021

PONE-D-20-33669

Knowledge and preventive practices towards COVID-19 among pregnant women seeking antenatal services in Ghana

PLOS ONE

Dear Author,

Thank you for submitting your manuscript to PLOS ONE. After careful consideration, we feel that it has merit but does not fully meet PLOS ONE’s publication criteria as it currently stands. Therefore, we invite you to submit a revised version of the manuscript that addresses the points raised during the review process.

We look forward to receiving your revised manuscript.

Kind regards,

Ramesh Kumar, PhD

Academic Editor

PLOS ONE

Journal Requirements:

Reviewers' comments:

Reviewer's Responses to Questions

**Comments to the Author**

1. Is the manuscript technically sound, and do the data support the conclusions?

Reviewer #1: Yes

Reviewer #2: Partly

2. Has the statistical analysis been performed appropriately and rigorously? 

Reviewer #1: Yes

Reviewer #2: No

3. Have the authors made all data underlying the findings in their manuscript fully available?

Reviewer #1: Yes

Reviewer #2: No

4. Is the manuscript presented in an intelligible fashion and written in standard English?

Reviewer #1: Yes

Reviewer #2: Yes

5. Review Comments to the Author

Reviewer #1: COVID-19 is a novel respiratory disease associated with severe morbidity and high mortality in the elderly population and people with comorbidities. Studies have suggested that pregnant women are more susceptible to COVID-19 compared to non-pregnant women. However, it’s unclear whether pregnant women in Ghana are knowledgeable about COVID-19 and practice preventive measures against it.

This manuscript I provides evidence of knowledge of covid -19 among pregnant women in Ghana. The Manuscript is technically sound, well written and data analysis supports the conclusion

Reviewer #2: The study was carried out in Ghana in the Nabdam district in the Northern part of the country. It sought to find out whether pregnant women we putting themselves at risk of COVID-19 in terms of following the stipulated COVID-19 preventive measures and also to find out whether they are in the known at all of these preventive measures.

Questionnaires were administered as the pregnant women visited the hospital for antenatal care. These are my concerns:

The northern part of Ghana in terms of demography and topology is totally different from the southern part so the title is misleading.

What informed in the selection of the particular health facilities. How many health facilities were selected and how dispersed were they. How can this be a representative study for the whole district. The scope of sampling is too small to generalize to the whole district. How is the attitude of the pregnant women in terms of attending antenatal.

The authors per the last paragraph states that no studies on knowledge and preventive practices are focused on pregnant women, but this is not so. So many studies on knowledge and preventive practices focused on pregnant women have been carried out by several people Nwafor et al 2020, Anikwe et al 2020, Fikadu et al 2021, Kassie et al 2021 among others. Authors should therefore state in clearer terms the novelty of the study.

Authors should check grammar , there are a lot of grammatical errors in the article

Authors should number the lines of the article to make it easier to make reference to.

The conclusion at the abstract section was just a result restated. Authors should conclude appropriately.

6. PLOS authors have the option to publish the peer review history of their article (what does this mean?). If published, this will include your full peer review and any attached files.

Reviewer #1: No

Reviewer #2: No

---

## [Author Response · Author response to Decision Letter 0]

19 Feb 2021

Reviewer #1 

COVID-19 is a novel respiratory disease associated with severe morbidity and high mortality in the elderly population and people with comorbidities. Studies have suggested that pregnant women are more susceptible to COVID-19 compared to non-pregnant women. However, it’s unclear whether pregnant women in Ghana are knowledgeable about COVID-19 and practice preventive measures against it.

This manuscript I provides evidence of knowledge of COVID -19 among pregnant women in Ghana. The Manuscript is technically sound, well written and data analysis supports the conclusion. 

Response: We sincerely appreciate the reviews.

Reviewer #2 

The study was carried out in Ghana in the Nabdam district in the Northern part of the country. It sought to find out whether pregnant women we putting themselves at risk of COVID-19 in terms of following the stipulated COVID-19 preventive measures and also to find out whether they are in the known at all of these preventive measures. Questionnaires were administered as the pregnant women visited the hospital for antenatal care. 

Response 

• We sincerely appreciate the reviews.

The northern part of Ghana in terms of demography and topology is totally different from the southern part so the title is misleading.

Response

• We have changed the title to reflect Northern Ghana. See lines 1 and 2. 

What informed in the selection of the particular health facilities? 

Response 

• All health facilities in the district that conducts antenatal care services were selected for the study. See lines 113 and 114. 

How many health facilities were selected? 

Response

• 16 health facilities were selected. See lines 113 and 114.

How dispersed were they? 

Response 

• The health facilities were not so dispersed. The distance between health facilities ranged from three to seven kilometers. 

How can this be a representative study for the whole district. The scope of sampling is too small to generalize to the whole district.

Response 

• Although the health facilities were purposively sampled, they included all antenatal care facilities in the district. Furthermore, pregnant women were randomly sampled from all these health facilities in the district. We therefore think that our findings can generalized in the district although same may not be applied in the country. We have stated it in our limitations. See lines 255, 256 and 257. 

How is the attitude of the pregnant women in terms of attending antenatal? 

Response 

• Our study did not assessed attitude of pregnant women towards antenatal care attendance. However, some of the authors are practitioners in the district and have observed that antenatal care attendance is high. 

The authors per the last paragraph states that no studies on knowledge and preventive practices are focused on pregnant women, but this is not so. So many studies on knowledge and preventive practices focused on pregnant women have been carried out by several people Nwafor et al 2020, Anikwe et al 2020, Fikadu et al 2021, Kassie et al 2021 among others. Authors should therefore state in clearer terms the novelty of the study?

Response 

• This study is the first of its kind in Ghana. We have appropriately stated that in the paper. See lines 97 and 98. 

Authors should check grammar, there are a lot of grammatical errors in the article.

Response

• We have thoroughly read the paper and grammatical errors have been corrected. 

Authors should number the lines of the article to make it easier to make reference to.

Response

• We have numbered all the lines of the article. 

The conclusion at the abstract section was just a result restated. Authors should conclude appropriately.

Response 

• The conclusion has been reconstructed appropriately. See lines 43,44 and 45.

---

## [Decision Letter · Decision Letter 1]

9 Apr 2021

PONE-D-20-33669R1

Knowledge and preventive practices towards COVID-19 among pregnant women seeking antenatal services in Northern Ghana

PLOS ONE

Dear Author,

Thank you for submitting your manuscript to PLOS ONE. After careful consideration, we feel that it has merit but does not fully meet PLOS ONE’s publication criteria as it currently stands. Therefore, we invite you to submit a revised version of the manuscript that addresses the points raised during the review process.

We look forward to receiving your revised manuscript.

Kind regards,

Ramesh Kumar, PhD

Academic Editor

PLOS ONE

Journal Requirements:

Additional Editor Comments (if provided):

Reviewers' comments:

Reviewer's Responses to Questions

**Comments to the Author**

1. If the authors have adequately addressed your comments raised in a previous round of review and you feel that this manuscript is now acceptable for publication, you may indicate that here to bypass the “Comments to the Author” section, enter your conflict of interest statement in the “Confidential to Editor” section, and submit your "Accept" recommendation.

Reviewer #1: All comments have been addressed

Reviewer #2: All comments have been addressed

2. Is the manuscript technically sound, and do the data support the conclusions?

Reviewer #1: Yes

Reviewer #2: Yes

3. Has the statistical analysis been performed appropriately and rigorously? 

Reviewer #1: Yes

Reviewer #2: Yes

4. Have the authors made all data underlying the findings in their manuscript fully available?

Reviewer #1: Yes

Reviewer #2: Yes

5. Is the manuscript presented in an intelligible fashion and written in standard English?

Reviewer #1: Yes

Reviewer #2: Yes

6. Review Comments to the Author

Reviewer #1: (No Response)

Reviewer #2: For the conclusion, state the type of education on COVID needed to be given pregnant women that is not being done already.

Abstract should emphasize on the Northern Ghana.

At the onset of the study there might not have been COVID-19 vaccines. Now there are, authors should kindly update this information in their introduction.

At the discussion section COVID-19 was mistakenly written as COVID-1, edit.

How does having at least primary education associate with easier access to health information compared to individuals without a formal education, explain further.

7. PLOS authors have the option to publish the peer review history of their article (what does this mean?). If published, this will include your full peer review and any attached files.

Reviewer #1: No

Reviewer #2: No

---

## [Author Response · Author response to Decision Letter 1]

9 Apr 2021

Reviewer #1: (No Response)

Response:

We sincerely appreciate your reviews. 

Reviewer #2: For the conclusion, state the type of education on COVID-19 needed to be given pregnant women that is not being done already. 

Response: Education of COVID-19 preventive practices at the health facility level has been emphasized. See page 3 line 44.

Abstract should emphasize on the Northern Ghana.

Response:

This has been edited. See page 2 line 27

At the onset of the study there might not have been COVID-19 vaccines. Now there are, authors should kindly update this information in their introduction.

Response: 

This information has been updated. See page 5 lines 84-86 and lines 91-92.

At the discussion section COVID-19 was mistakenly written as COVID-1, edit.

Response: 

This has been edited. See page 12 line 204

How does having at least primary education associate with easier access to health information compared to individuals without a formal education, explain further.

Response: 

It exposes mothers to knowledge on health compared to mothers with no formal education. We have cited this sentence in page 13 line 221.

---

## [Decision Letter · Decision Letter 2]

26 Apr 2021

PONE-D-20-33669R2

Knowledge and preventive practices towards COVID-19 among pregnant women seeking antenatal services in Northern Ghana

PLOS ONE

Dear Author,

Thank you for submitting your manuscript to PLOS ONE. After careful consideration, we feel that it has merit but does not fully meet PLOS ONE’s publication criteria as it currently stands. Therefore, we invite you to submit a revised version of the manuscript that addresses the points raised during the review process.

We look forward to receiving your revised manuscript.

Kind regards,

Ramesh Kumar, PhD

Academic Editor

PLOS ONE

Reviewers' comments:

Reviewer's Responses to Questions

**Comments to the Author**

1. If the authors have adequately addressed your comments raised in a previous round of review and you feel that this manuscript is now acceptable for publication, you may indicate that here to bypass the “Comments to the Author” section, enter your conflict of interest statement in the “Confidential to Editor” section, and submit your "Accept" recommendation.

Reviewer #2: All comments have been addressed

Reviewer #3: All comments have been addressed

2. Is the manuscript technically sound, and do the data support the conclusions?

Reviewer #2: Yes

Reviewer #3: Partly

3. Has the statistical analysis been performed appropriately and rigorously? 

Reviewer #2: (No Response)

Reviewer #3: Yes

4. Have the authors made all data underlying the findings in their manuscript fully available?

Reviewer #2: Yes

Reviewer #3: No

5. Is the manuscript presented in an intelligible fashion and written in standard English?

Reviewer #2: Yes

Reviewer #3: Yes

6. Review Comments to the Author

Reviewer #2: (No Response)

Reviewer #3: This study aimed to assess the knowledge and preventive practices towards COVID-19 among pregnant women seeking antenatal services in Northern Ghana. A cross-sectional study was conducted using a structured questionnaire in the Nabdam district in Ghana. A total of 527 pregnant women were randomly sampled from health facilities offering antenatal care services in the district. Multivariable logistic regression analysis was used to assess the association between the predictors and outcome variables. Although the majority of the women had adequate knowledge of COVID-19, less than half of them were engaged in good COVID-19 preventive practices. Education of pregnant women on COVID-19 preventive practices should be intensified at health facilities while improving upon the water, sanitation and hygiene need particularly in rural communities.

This study has merit and important in the context of Ghana. However, I have several comments on the content.

Major Comments:

1. “The sample size for the study was estimated using EpiInfo Version 7.1 (STAT CALC). The prevalence of knowledge on COVID-19 was not known, therefore a 50% prevalence rate was used with a 95% confidence interval and 5% margin of error. The estimated minimum sample size was 407 including a 10% non-response rate. However, 527 participants completed our surveys.” Why was there oversampling and how this might affect the study?

2. Line 139-140: “A score of 3 points and above was considered good COVID-19 preventive practices while 2 points and below was considered poor COVID-19 preventive practices.” Was this cut-off validated? Please provide reference of previous literature who used this score.

Minor Comments:

1. There are some typos in the manuscript. For example in line 90 “Ghana reported its first case of COVIG-19 on 12th March 2020 in its national capital, Accra [21].” Please change this to COVID-19.

2. Line 121: add the unit (years) after the age.

7. PLOS authors have the option to publish the peer review history of their article (what does this mean?). If published, this will include your full peer review and any attached files.

Reviewer #2: No

Reviewer #3: **Yes: **Rajat Das Gupta

---

## [Author Response · Author response to Decision Letter 2]

27 Apr 2021

Reviewer #3: 

This study aimed to assess the knowledge and preventive practices towards COVID-19 among pregnant women seeking antenatal services in Northern Ghana. A cross-sectional study was conducted using a structured questionnaire in the Nabdam district in Ghana. A total of 527 pregnant women were randomly sampled from health facilities offering antenatal care services in the district. Multivariable logistic regression analysis was used to assess the association between the predictors and outcome variables. Although the majority of the women had adequate knowledge of COVID-19, less than half of them were engaged in good COVID-19 preventive practices. Education of pregnant women on COVID-19 preventive practices should be intensified at health facilities while improving upon the water, sanitation and hygiene need particularly in rural communities.

This study has merit and important in the context of Ghana. However, I have several comments on the content.

Response:

We sincerely appreciate your reviews. 

Major Comments:

1. “The sample size for the study was estimated using EpiInfo Version 7.1 (STAT CALC). The prevalence of knowledge on COVID-19 was not known, therefore a 50% prevalence rate was used with a 95% confidence interval and 5% margin of error. The estimated minimum sample size was 407 including a 10% non-response rate. However, 527 participants completed our surveys.” Why was there oversampling and how this might affect the study?

Response:

The sample size of 407 was the minimum sample size required. A sample size of 527 will increase the precision of our estimates. 

2. Line 139-140: “A score of 3 points and above was considered good COVID-19 preventive practices while 2 points and below was considered poor COVID-19 preventive practices.” Was this cut-off validated? Please provide reference of previous literature who used this score.

Response: 

This cut-off was adapted from Bloom’s cut-off points. It was duly cited in our study. See page 7 line 139.

Minor Comments:

1. There are some typos in the manuscript. For example, in line 90 “Ghana reported its first case of COVIG-19 on 12th March 2020 in its national capital, Accra [21].” Please change this to COVID-19.

Response: This error has been corrected. See page 5 line 90. We have also read through the manuscript and made all the necessary correction of errors. 

2. Line 121: add the unit (years) after the age.

Response: 

Years has added accordingly. See page 6 lines 121-122.

---

## [Decision Letter · Decision Letter 3]

7 Jun 2021

Knowledge and preventive practices towards COVID-19 among pregnant women seeking antenatal services in Northern Ghana

PONE-D-20-33669R3

Dear Author,

We’re pleased to inform you that your manuscript has been judged scientifically suitable for publication and will be formally accepted for publication once it meets all outstanding technical requirements.

Kind regards,

Ramesh Kumar, PhD

Academic Editor

PLOS ONE

Additional Editor Comments (optional):

Reviewers' comments:

Reviewer's Responses to Questions

**Comments to the Author**

1. If the authors have adequately addressed your comments raised in a previous round of review and you feel that this manuscript is now acceptable for publication, you may indicate that here to bypass the “Comments to the Author” section, enter your conflict of interest statement in the “Confidential to Editor” section, and submit your "Accept" recommendation.

Reviewer #2: All comments have been addressed

Reviewer #3: All comments have been addressed

2. Is the manuscript technically sound, and do the data support the conclusions?

Reviewer #2: Yes

Reviewer #3: Yes

3. Has the statistical analysis been performed appropriately and rigorously? 

Reviewer #2: Yes

Reviewer #3: Yes

4. Have the authors made all data underlying the findings in their manuscript fully available?

Reviewer #2: Yes

Reviewer #3: Yes

5. Is the manuscript presented in an intelligible fashion and written in standard English?

Reviewer #2: Yes

Reviewer #3: Yes

6. Review Comments to the Author

Reviewer #2: (No Response)

Reviewer #3: The authors have addressed all the comments. I recommend this manuscript for publication. I wish the authors good luck.

7. PLOS authors have the option to publish the peer review history of their article (what does this mean?). If published, this will include your full peer review and any attached files.

Reviewer #2: No

Reviewer #3: **Yes: **Rajat Das Gupta

---

## [Editor Report · Acceptance letter]

9 Jun 2021

PONE-D-20-33669R3 

Knowledge and preventive practices towards COVID-19 among pregnant women seeking antenatal services in Northern Ghana 

Dear Dr. Kumbeni:

I'm pleased to inform you that your manuscript has been deemed suitable for publication in PLOS ONE. Congratulations! Your manuscript is now with our production department. 

Kind regards, 

on behalf of

Dr. Ramesh Kumar 

Academic Editor

PLOS ONE